# Describing fine spatiotemporal dynamics of rat fleas in an insular ecosystem enlightens abiotic drivers of murine typhus incidence in humans

Annelise Tran[1,2,3,4☯]*, Gildas Le Minter[5☯], Elsa Balleydier[6], Anaïs Etheves[3,4], Morgane Laval[3,4], Floriane Boucher[3,4¤a], Vanina Guernier[5¤b], Erwan Lagadec[5¤c], Patrick Mavingui[5], Eric Cardinale[3,4], Pablo Tortosa[5]

1 CIRAD, UMR TETIS, Sainte-Clotilde, Reunion Island, France, 2 TETIS, Univ Montpellier, AgroParisTech, CIRAD, CNRS, INRAE, Montpellier, France, 3 CIRAD, UMR ASTRE, Sainte-Clotilde, Reunion Island, France, 4 ASTRE, Univ Montpellier, CIRAD, INRAE, Montpellier, France, 5 UMR PIMIT, Univ La Réunion, INSERM, CNRS, IRD, CYROI, Sainte-Clotilde, Reunion Island, France, 6 Santé Publique France, cellule régionale, French Public Health Agency, Regional Unit, Saint Denis, Reunion Island, France

☯ These authors contributed equally to this work.
¤a Current address: GDS France, Vitry-sur-Seine, France
¤b Current address: National Animal Disease Center, Agricultural Research Service, United States Department of Agriculture, Ames, Iowa, USA; Oak Ridge Institute for Science and Education, Agricultural Research Service Research Participation Program, Oak Ridge, Tennessee, United States of America
¤c Current address: Fish Disease Research Group, Department of Biological Sciences, University of Bergen, Bergen, Norway
* annelise.tran@cirad.fr

**Data Availability Statement:** All relevant data are within the paper and its Supporting Information files. Predicted Xenopsylla Genus Flea Index map is

## Abstract

Murine typhus is a flea-borne zoonotic disease that has been recently reported on Reunion Island, an oceanic volcanic island located in the Indian Ocean. Five years of survey implemented by the regional public health services have highlighted a strong temporal and spatial structure of the disease in humans, with cases mainly reported during the humid season and restricted to the dry southern and western portions of the island. We explored the environmental component of this zoonosis in an attempt to decipher the drivers of disease transmission. To do so, we used data from a previously published study (599 small mammals and 175 *Xenopsylla* fleas from 29 sampling sites) in order to model the spatial distribution of rat fleas throughout the island. In addition, we carried out a longitudinal sampling of rats and their ectoparasites over a 12 months period in six study sites (564 rats and 496 *Xenopsylla* fleas) in order to model the temporal dynamics of flea infestation of rats. Generalized Linear Models and Support Vector Machine classifiers were developed to model the *Xenopsylla* Genus Flea Index (GFI) from climatic and environmental variables. Results showed that the spatial distribution and the temporal dynamics of fleas, estimated through the GFI variations, are both strongly controlled by abiotic factors: rainfall, temperature and land cover. The models allowed linking flea abundance trends with murine typhus incidence rates. Flea infestation in rats peaked at the end of the dry season, corresponding to hot and dry conditions, before dropping sharply. This peak of maximal flea abundance preceded the annual peak of human murine typhus cases by a few weeks. Altogether, presented data raise novel

available in the following public repository: CIRAD Dataverse, https://dataverse.cirad.fr/dataset.xhtml?persistentId=doi:10.18167/DVN1/TWNWG6.

**Funding:** This study was funded by the Regional Health Agency in Reunion Island (https://www.ocean-indien.ars.sante.fr/), FEDER INTERREG TROI project, and FEDER-POCT LeptOI project, under the platform in partnership One Health Indian Ocean (www.onehealth-oi.org). The funders had no role in study design, data collection and analysis, decision to publish, or preparation of the manuscript.

**Competing interests:** The authors have declared that no competing interests exist.

questions regarding the ecology of rat fleas while developed models contribute to the design of control measures adapted to each micro region of the island with the aim of lowering the incidence of flea-borne diseases.

## Author summary

Murine typhus is a neglected zoonotic disease, as the number of human cases is likely underestimated in the absence of specific symptoms. It is caused by *Rickettsia typhi*, a pathogenic bacterium transmitted by rat fleas (*Xenospylla* spp). The distribution and dynamics of this disease result from complex interactions involving vectors, reservoirs and humans within a shared environment. In this study, we explored the environmental drivers of rat fleas' abundance on Reunion Island, where murine typhus has recently emerged. Results showed that *i)* rat fleas' abundance is highly dynamic, characterized by a peak at the end of the dry season and *ii)* among the factors investigated, rainfall, temperature and land cover are the main determinants of rat fleas' abundance. We modeled a predictive map of flea distribution that strongly correlates with the spatial distribution of human cases on the island. This study highlights the importance of accounting for environmental and climatic characteristics to better understand the spatial and temporal drivers of flea-borne diseases.

## Introduction

Zoonotic pathogens are defined as infectious agents transmitted among vertebrate animals and humans [1,2]. In a number of zoonoses of major medical concern, humans are only incident hosts unable to maintain persistent secondary transmission of the pathogens and hence playing an anecdotal role in the evolution and biological cycles of these microorganisms. For such diseases, investigations aiming at deciphering environmental transmission cycles are crucial to determine conditions favoring spillover transmission from animal to human populations. The first step of such investigations relies on the identification of animal reservoirs. Subsequently, it is paramount to thoroughly describe the spatiotemporal dynamics of zoonotic pathogens *in natura* and the biotic or abiotic factors controlling these dynamics. Such a comprehensive knowledge is likely to significantly improve the understanding of the ecology of a given zoonotic pathogen, which in turn will orientate the design of control measures towards enhanced effectiveness.

Spatiotemporal dynamics of transmission can be characterized at different scales, from global, regional to local scales. Implementing high throughput surveillance systems able to detect above normal incidence of zoonoses at a global scale allows pinpointing emergence episodes and implementing early diagnosis. However, the ecology of a given pathogen is likely tightly associated to its niche, and identifying the drivers of the maintenance and emergence of a zoonotic disease requires the description of fine spatiotemporal dynamics of the disease within its discrete environmental setup.

Such investigations are obviously challenging given the number of animal species potentially acting as reservoirs or vectors, and are further made more complex by migrating taxa. Insular ecosystems, and especially remote oceanic islands of limited size, are ideal settings for such investigations due to a limited biodiversity as well as reduced migration [3]. Reunion Island, a geographically isolated volcanic island located 700 km East of Madagascar has

experienced in the last decade the emergence of several vector borne diseases [4,5], including murine typhus, caused by the bacterium *Rickettsia typhi* [6]. The detection of the emergence of this disease in the island may have been delayed due to suboptimal surveillance and diagnosis prior the identification of the first human cases in 2012 [7]. Indeed, non-specific symptoms as well as low prevalence keep diseases such as murine typhus under-reported from even sophisticated surveillance systems [8]. Despite a low incidence (in average, 8.8 cases have been detected each year between 2011 and 2019), the epidemiology of murine typhus on Reunion Island displays some interesting patterns: a strong geographic structure, cases being mostly restricted to the western and southern dry portions of the island [6,7]; and a strong seasonality with cases peaking during the warm rainy season, between December and February [6].

Rats are considered a major reservoir of *R. typhi* [9]. These rodents are widespread throughout the island as is typical on tropical islands [10,11]. Therefore, it is likely that their distribution does not play a significant role in the geographic structure of the disease. By contrast, such structure may be driven by the distribution of fleas (hypothesis H1), which can be verified by gathering flea infestation indices of rats trapped in distinct habitats of the island. This hypothesis is indeed substantiated by a previous study reporting that rat fleas are virtually absent from the eastern humid windward coast while they are abundant in the dry savannah of the western coast [12]. As far as temporal dynamics are concerned, the seasonality of murine typhus may result from marked variations in either the abundance of fleas (H2) or *R. typhi* infection prevalence in fleas (H3) throughout the year, or from transient increases in human exposure to rats (H4), as documented in Madagascar where plague seasonality is associated with rats' movements from the field to the villages, likely directed by the availability of food [13].

The identification of the environmental drivers of the strong geographic structure and seasonality of murine typhus in Reunion Island is the focus of the present investigation. We tested two of the four previously stated hypotheses by using sampling data collected in distinct habitats and throughout all seasons. We developed a predictive map of flea distribution and tested whether the geographical structure of flea distribution and human cases are congruent (H1). In addition, we conducted a temporal investigation of rats' infestation by fleas in order to test whether the seasonality of human disease results from seasonality in flea abundance (H2). Finally, we used gathered data and resulting conclusions to propose the implementation of simple control measures aimed at lowering murine typhus incidence on the island.

## Methods

### Ethics statements

The ethical terms of the research protocol were approved by the CYROI institutional ethical committee (Comité d'Ethique du CYROI n˚114, certified by Ministry of Higher Education and Research) and received the agreement number 03387. All animal procedures carried out in our study were performed in accordance with the European Union legislation for the protection of animals used for scientific purposes (Directive 2010/63/EU).

For the purpose of population health surveillance, local health authorities gave the regional team of Santé Publique France access to health data related to each case. This surveillance was conducted according to the authorization 314194-V42 of the French Data protection Authority (French acronym CNIL) related to urgent investigation. The prospective study was run in accordance with authorization 14.762 from the Advisory Committee on Information Processing in Material Research in the Field of Health (French acronym CCPPRB) and with the 915307 authorization from CNIL.

## Study area

Reunion Island is a small, densely populated island (2,512 km², 866,506 inhabitants in 2019) located in the southwestern Indian Ocean (Fig 1). The climate is subtropical with two main seasons: a dry and mild season from May to October, and a rainy and hot season from November to April. This latter season is under the influence of tropical lows (including cyclones) associated with heavy rainfalls. Because of the mountainous and rugged relief, the climate is highly contrasted: the east windward coast is notoriously humid while the west leeward coast is distinctly drier. Of note, temperatures decrease progressively from the coast to the central mountains, with marked differences due to the elevations of mountainous central landscapes (>3,000 meters).

## Rats and fleas data

Data used for the analyses included samples collected in the frame of a previously published study [12] as well as a 12-months sampling survey.

**Dataset 1.** Data from rodent trapping conducted in 2012–2013 in different biotopes [12] were used to model the spatial distribution of flea infestation. Small mammals were trapped in different sites from November 2012 to November 2013 (each site was sampled once or twice), fleas collected and subsequently morphologically and molecularly identified (see detailed description in [12]). A selection was made on the trapping results corresponding to the hot wet season (November to May), when a greater abundance of fleas was observed [12]. In the

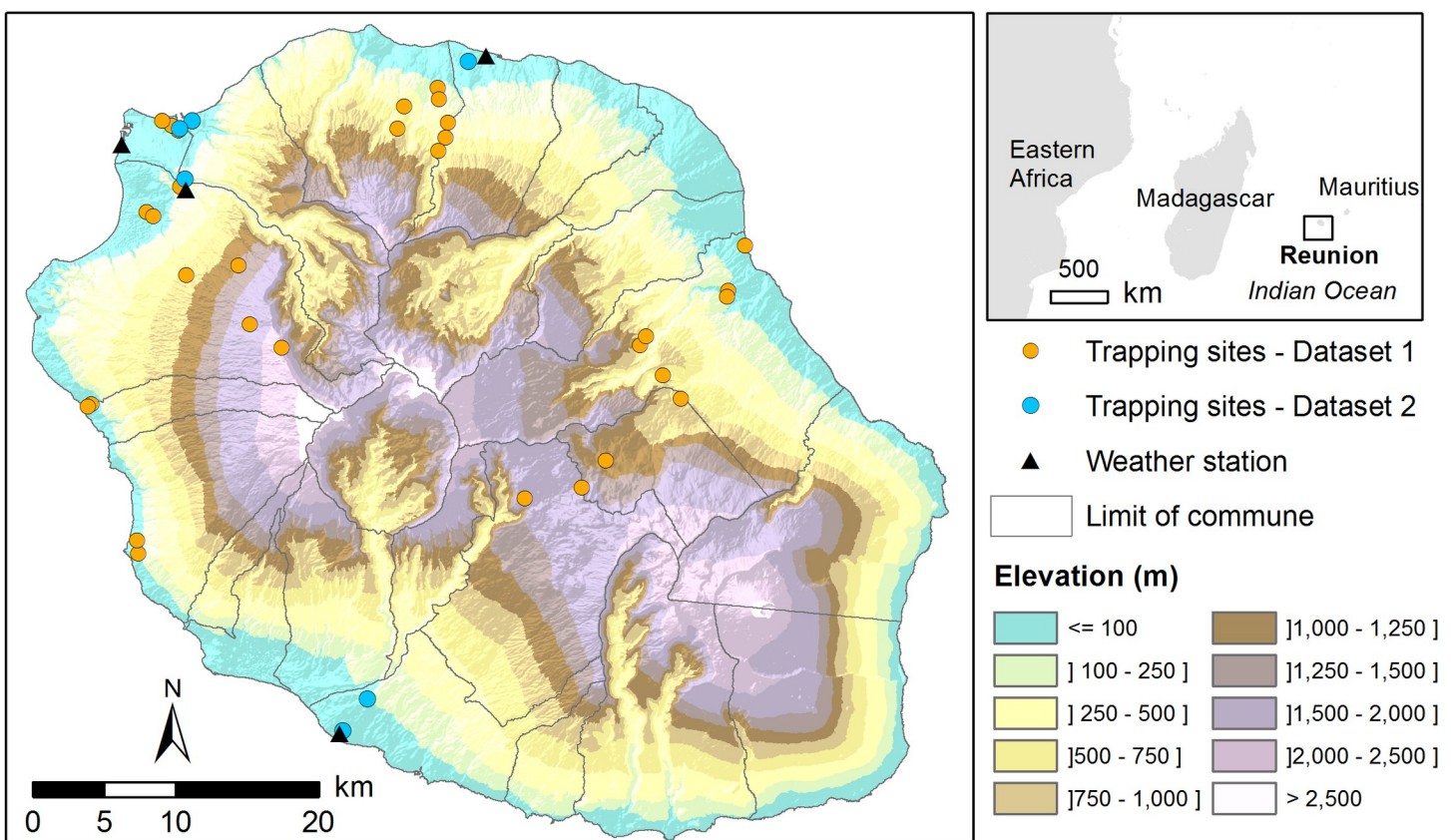

**Fig 1. Location of the study area.** The map shows the relief of Reunion Island (source: French national geographic institute, BD ALTI–data available under the free license Open license 2.0) as well as sampling sites for each dataset and the location of the weather stations used for modelling.

corresponding 29 sampling sites (Fig 1 and S1 and S2 Tables), a total of 599 small mammals were trapped and 175 *Xenopsylla sp.* fleas sampled during the hot wet season.

**Dataset 2.** Data from a longitudinal trapping survey conducted in 2017–2018 in six study sites (Fig 1) were used to model the seasonal dynamics of flea infestation on rats. Small mammal hosts were trapped every 8 to 10 weeks in each site from March 2017 to February 2018 (detailed description in Supplementary Information S1 Text and S1 and S3 Tables). A total of 564 small mammals and 496 *Xenopsylla sp.* fleas were sampled.

## Epidemiological data

Confirmed murine typhus cases in humans were recorded by the regional cell of Santé Publique France in La Reunion between 2011 and 2018, and defined as one of the following:

- $> = 4$-fold increase in antibody response against *R. typhi* antigen from acute- and convalescent-phase serum samples by positive indirect immunofluorescent assays (IFA) for IgM or IgG antibodies;

- A positive specific Polymerase Chain Reaction (PCR) result;

- A single IgM or IgG level of $> = 1{:}1024$ by IFA;

- A seroconversion (from a first negative serology to a second positive);

- A positive serology confirmed by Western blot by the Rickettsia National Reference Center (Marseille, France).

We recorded 81 confirmed murine typhus cases, with location information available for all but one. Most of the cases (67.9%) were located in the south of the island, 30.9% in the west and 1.2% in the north. No cases were reported in the east. Assuming that cases were infected in their neighborhood, the confirmed murine typhus cases for which the location was known (80 cases), were linked to their IRIS ("Ilots Regroupés pour l'Information Statistique", n = 344) level, corresponding to the most detailed population census area in Reunion (Table 1). For IRIS areas including more than one confirmed murine typhus case (n = 47 IRIS), the incidence

**Table 1. Sources of geographic data.**

| Data | Description | Data source |
|---|---|---|
| **Annual minimum temperature map** | Average minimum annual temperature of 73 weather stations (reference period: 1987–2017) | CIRAD / Meteo-France, available at http://aware.cirad.fr/layers/geonode%3Atemp_min |
| **Annual maximum temperature map** | Average maximum annual temperature of 73 weather stations (reference period: 1987–2017) | CIRAD / Meteo-France, available at http://aware.cirad.fr/layers/geonode%3Atemp_max |
| **Annual mean temperature map** | Average mean annual temperature of 73 weather stations (reference period: 1987–2017) | CIRAD / Meteo-France, available at http://aware.cirad.fr/layers/geonode%3Atemp_moy |
| **Annual median rainfall map** | Median annual rainfall observed in Reunion between 1986 and 2016 from 143 rain gauges | CIRAD / Meteo-France, available at http://aware.cirad.fr/layers/geonode%3Apluie_median_1986_2016 |
| **Land cover map** | Object-oriented remote sensing product developed from SPOT 5 images fused at 2.5 meters resolution | Institut de Recherche pour le Développement, UMR Espace-Dev, available at http://homisland.seas-oi.org/ |
| **Human population** | IRIS ("Ilots regroupés pour l'information statistique") census areas | Institut Géographique National (IGN), Contour IRIS édition 2019, available at ftp://Contours_IRIS_ext:ao6Phu5ohJ4jaeji@ftp3.ign.fr/CONTOURS-IRIS_2-1__SHP__FRA_2019-01-01.7z.001 |
| | Number of inhabitants per IRIS (2016 census data) | Institut National de la Statistique et des Etudes Economiques (INSEE), available at https://www.insee.fr/fr/statistiques/4228434 |

rate (number of confirmed cases between 2011 and 2018 per 10,000 inhabitants) per IRIS was calculated using 2016 census data.

## Meteorological and environmental data

**Daily weather data 2017–2018.** We used the meteorological service 'Meteo France' (https://publitheque.meteo.fr) providing the daily temperature (minimum, maximum, and mean) and rainfall records from 2016 to 2018. Four weather stations were selected, being the closest to the six study sites of Dataset 2 where the longitudinal trapping survey was conducted (Fig 1).

**Long-term temperature and rainfall data.** Maps of annual minimum, maximum and mean temperatures, as well as annual median rainfall were downloaded from the web portal aware.cirad.fr (Table 1).

**Land cover map.** A land cover map of Reunion Island derived from SPOT-5 satellite images was downloaded from the web portal homisland.seas-oi.org (Table 1). It includes 10 land cover classes: pasture, sugar cane, heterogeneous cropland, continuous urban, discontinuous urban, forest, shrub vegetation, herbaceous vegetation, barren land, and water [14,15].

## Modelling the spatial distribution of rat fleas

*Xenopsylla cheopis* and *Xenopsylla brasiliensis* are known vectors of *R. typhi* [16], both present on the island, but the two species are difficult to distinguish using morphological characters only, especially in females. Therefore, we defined the Genus Flea Index (GFI) as the total number of collected fleas belonging to the genus *Xenopsylla* divided by the total number of trapped mammals. GFI, representing the mean number of *Xenopsylla* spp. fleas sampled per trapped mammal, was calculated for each trapping site of Dataset 1 (n = 29) (S2 Table).

GFI was analyzed in relationship to ecological and climatic variables using uni-variable and multi-variable generalized linear modelling to investigate the spatial association between the GFI (variable to explain) and the following explanatory variables: annual minimum, maximum and mean temperature, annual median rainfall, percentage of different land cover types (agriculture, continuous urban, discontinuous urban, forest, shrub vegetation, herbaceous vegetation, barren land, water) in a 200-meters radius buffer. This distance corresponds to the maximum trip length for rats estimated in two tropical islands of southwestern Indian Ocean using mark-recapture sessions [17]. The explanatory variables were calculated for each trapping site of Dataset1 using Geographic Information System spatial analysis functions (ESRI ArcGIS Spatial Analyst).

Poisson distribution function was used to fit the models, as it is appropriate for the analysis of count data. All variables were first tested in univariate analysis, and significant variables at a 0.05 $p$-value were selected. To reduce multicollinearity in the statistical model, pair-wise correlations between significant explanatory variables were assessed. Within a group of correlated variables (pairwise correlation coefficients greater than 0.7 or less than -0.7), only the most strongly associated variable (based on the likelihood ratio test) was further used for multivariate modelling. The final multivariate model was developed by backward elimination regression modelling. It was used to predict values of GFI for the entire island by applying the model to the climatic and environmental datasets using GIS (ESRI ArcGIS Spatial Analyst). Predicted GFI values were extracted at the location of each trapping site of Dataset1 and Dataset2, for comparison with observed GFI values using the root mean square error (RMSE).

## Modelling the temporal dynamics of rat fleas

Data from a longitudinal trapping survey (Dataset2) were used to model the temporal dynamics of rat fleas. GFI was calculated for each trapping site of Dataset 2 (n = 6) and each date of

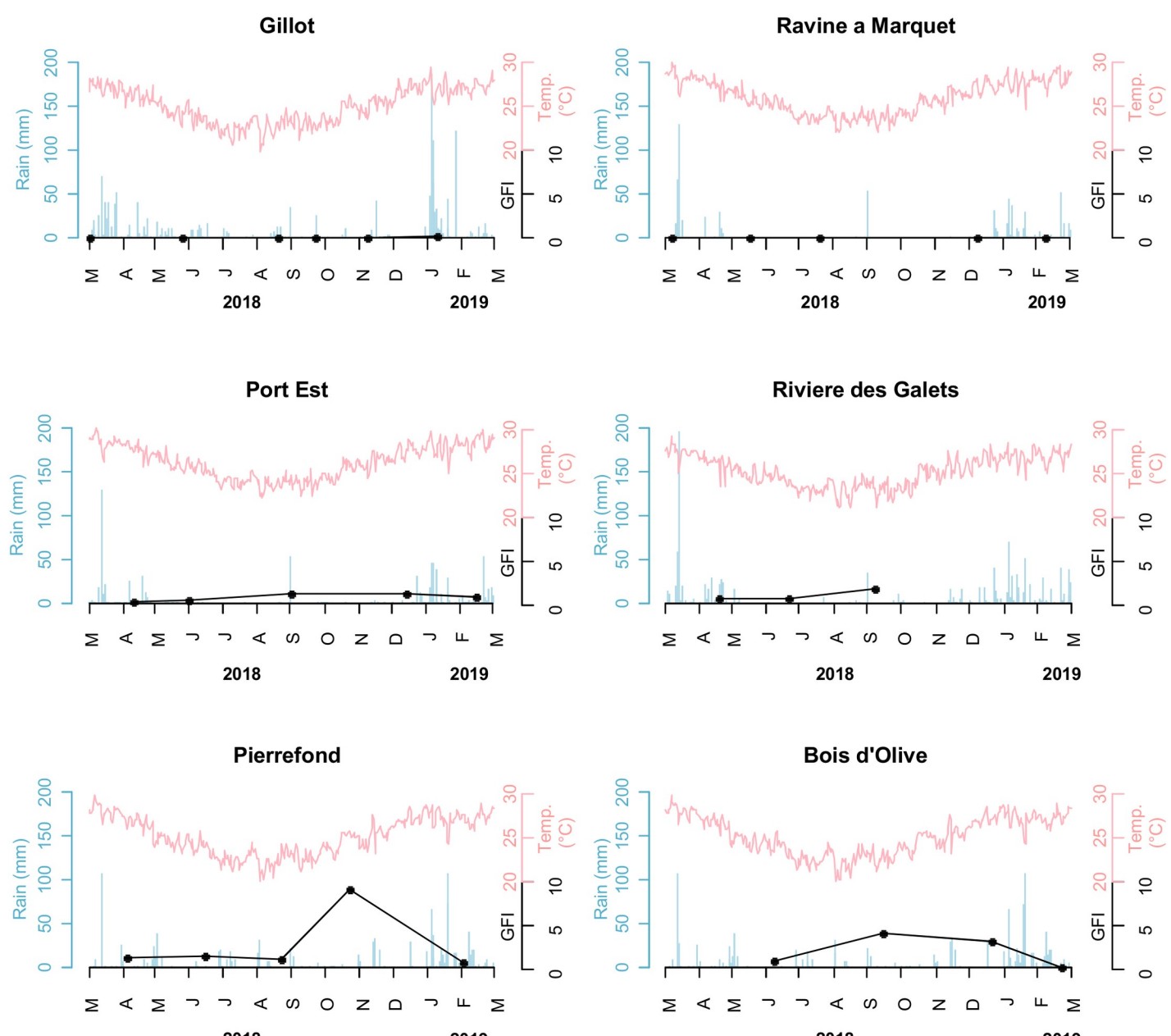

**Fig 2. Dynamics of *Xenopsylla* genus flea index in six study sites, Reunion Island, 2018–2019.** The GFI, rainfall and temperature are plotted in black, blue and red colors, respectively.

trapping from March 2017 to February 2018 (Fig 2 and S3 Table). For the sites with annual average GFI greater than one (Pierrefond and Bois d'Olive), the temporal dynamics of GFI in relationship to meteorological variables was modelled using Support Vector Machine (SVM) supervised learning technique. The following variables were calculated with a retroactive period (last $N$ days) starting at the capture date: minimum, maximum and average temperatures, rain accumulation and maximum rainfall ($N \in [0, 7, 14, 21, \ldots 63]$). The SVM method allowed accounting for non-linear relationships to separate high from low GFI values from a combination of at most three input meteorological variables to avoid overfitting. It was used

with a radial kernel, a cost coefficient set to 1 and the epsilon parameter set to 1. All of the statistical calculations were performed in R language [18] using the package "e1071" [19]. The model with the best performance was identified using a selection criterion based on the mean square error.

### Modelling the risk of murine typhus transmission

For each IRIS area with more than one confirmed murine typhus case (n = 47), the mean *Xenopsylla* GFI value was computed (software: ESRI ArcMap and ArcMap Spatial Analyst Extension, Redlands, CA, USA) and compared to the incidence rate using Pearson correlation coefficient and linear regression.

## Results

### Rainfall, temperature and land cover explain the spatial heterogeneity of rat fleas

Univariate analysis identified ten variables significantly associated ($p<0.01$) with the GFI (S4 Table), narrowed down to six variables in the multivariate regression analysis (Table 2). The infestation of rats by *Xenopsylla* spp. fleas, as captured by the GFI, was found negatively associated with the annual median rainfall, and positively associated with the annual maximum temperature and the surface percentage within a 200-m buffer of the following land cover classes: "discontinuous urban", "forest", "herbaceous vegetation" and "agriculture". The map of the predicted *Xenopsylla* GFI based on the climatic and environmental variables (Fig 3A) highlighted several areas with high predicted GFI values, mainly located in the western coast. Fig 3B shows the predicted values for *Xenopsylla* GFI compared to the observations. RMSEs were 1.16 and 2.17 for Dataset1 and Dataset 2 respectively.

### Precipitation and temperature explain the temporal dynamics of rat fleas

In the sites recording high levels of infestation (Fig 2), GFI peaked in November in Pierrefond (South of the Island) and in September in Port-Est and Bois d'Olive, located in the West and South of the island, respectively. Then GFI sharply dropped at all three sites within the two following months.

The model with the best performance (S5 Table) included a set of two variables: the maximum temperature and the maximum rainfall over the last 42 days (Fig 4). The favorable conditions for high GFI are a daily maximum temperature above 27˚C and a maximum rainfall over the last 42 days below 60 mm. These hot and dry conditions correspond to the end of the dry season (September–December months, depending on the year and location) on Reunion Island.

**Table 2. Multivariate model explaining *Xenopsylla* Genus Flea Index from climatic and environmental variables.**

| Variable | Estimate | p-value |
|---|---|---|
| Intercept | -5.558 | |
| Annual median rainfall (mm) | $-4.487 \ 10^{-3}$ | $< 10^{-15}$ |
| Annual maximum temperature (˚C) | $3.494 \ 10^{-1}$ | $< 10^{-7}$ |
| Discontinuous urban | 16.598 | $< 10^{-15}$ |
| Forest | 3.903 | $< 10^{-15}$ |
| Herbaceous vegetation | 8.243 | $< 10^{-15}$ |
| Agriculture | 4.392 | $< 10^{-15}$ |

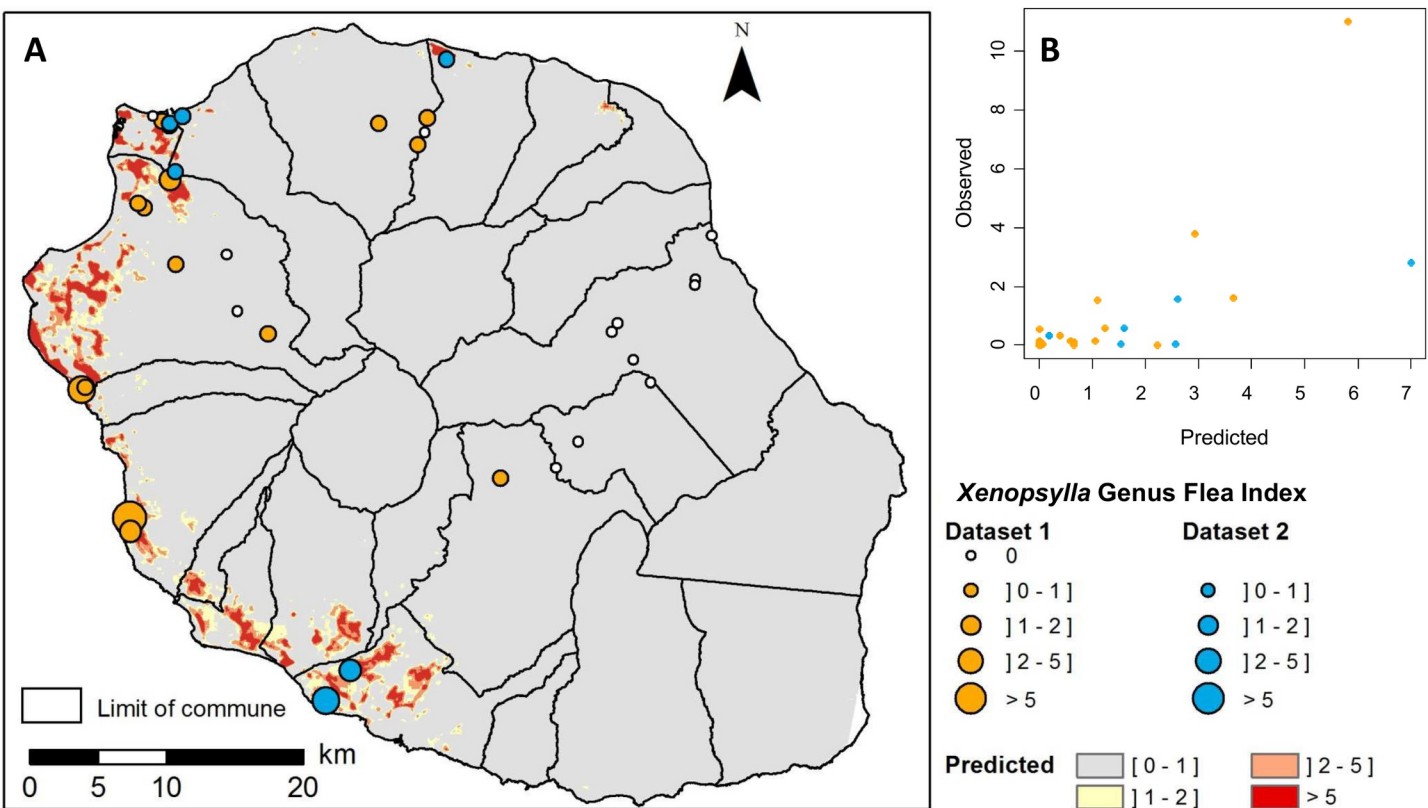

**Fig 3. Observed and predicted *Xenopsylla* Genus Flea Index (GFI) in Reunion Island. A) Map of predicted GFI based on climatic and environmental variables** (source for the administrative limits: French national geographic institute BD TOPO–data available under the free license Open license 2.0) **B) Bi-dimensional representation of predicted and observed GFI values.** Observed GFI values from dataset 1 and dataset 2 are plotted in orange and blue colors, respectively.

According to our model, the predicted GFI values steadily increase through the year from February to December (Fig 5). Predicted GFI peaks in December, corresponding to the annual increase of murine typhus cases in humans (Fig 5). Then, predicted GFI rapidly decreases to reach its minimum in February, while the number of human cases remains high and drops with a time lag of two months after the GFI decrease.

## The predicted spatial distribution of *Xenopsylla* fleas is significantly correlated with murine typhus incidence

A strong concordance was observed between the predicted GFI map (Fig 3A) and the murine typhus incidence map (Fig 6A) on Reunion Island. At IRIS level, mean GFI and murine typhus incidence rates were significantly correlated (Pearson correlation coefficient r = 0.47, p<0.05). Fig 6B shows the predicted values of the GFI compared to the murine typhus incidence at IRIS level.

## Discussion

Understanding and predicting the transmission dynamics of vector-borne pathogens such as *R. typhi* is challenging as such dynamics result from complex interactions involving vectors, reservoirs and humans within a shared environment. In Reunion Island, non-specific symptoms of murine typhus associated with a recent emergence of the disease and a lack of awareness of family doctors are expected to lead to an underestimation of the number of cases,

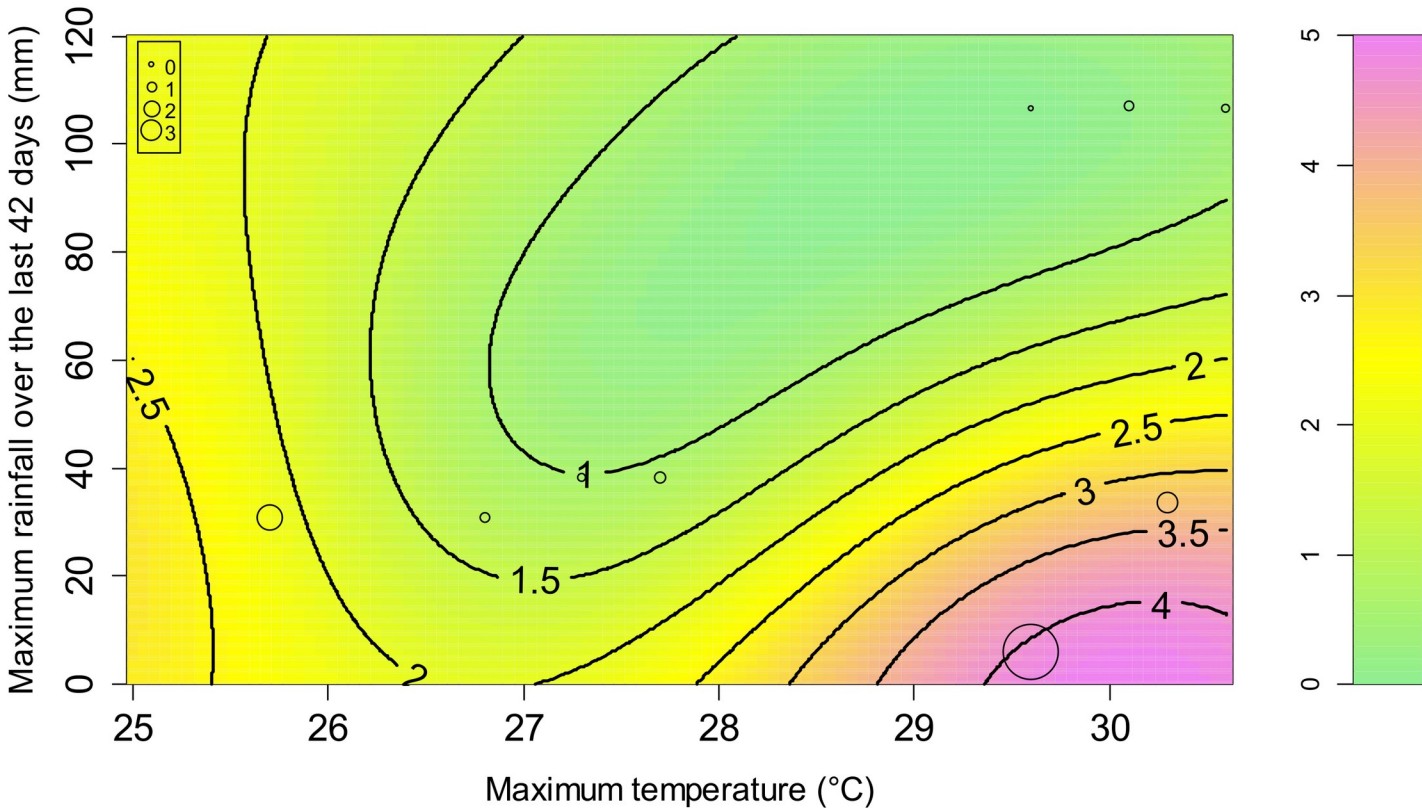

**Fig 4. Prediction of the *Xenopsylla* Genus Flea Index (GFI) according to two variables related to two meteorological variables: the maximum temperature and the maximum rainfall over the last 42 days.** The colors and level lines represent the model predictions. The circles correspond to the observations.

which makes it difficult to study disease risk factors using only statistical analyses of the human cases. In the present study, we analyzed factors related to rodents and rodent fleas compartments, both key components of *R. typhi* maintenance and transmission. The results of our analysis demonstrate that climatic and environmental variables strongly affect the dynamics and spatial distribution of adult rat fleas, and may at least in part explain the spatial and temporal occurrence of the human disease.

## Spatial patterns of flea abundance

We used accessible data [12] to identify the climatic and environmental drivers of the spatial distribution of *Xenopsylla* rat fleas; these drivers were then used to produce a predictive map of flea abundance on Reunion Island. Our analysis showed that low precipitation and high temperatures favor flea abundance. The effect of temperature and humidity on fleas' development has been experimentally measured in a handful of species. A recent experimental study showed that the development of *X. cheopis* and *Synopsyllus fonquerniei*, both known vectors of plague in Madagascar, was shortened by elevated temperatures while no effect of humidity was noticed in the used experimental conditions [20]. Krasnov and colleagues (2001) showed that egg hatch as well as larval development of *Xenopsylla conformis* and *Xenopsylla ramesis*, both parasites of the same rodent species, were strongly affected by temperature and humidity [21]. In addition, an examination of the rodent burrows in different habitats revealed large differences in burrow architecture possibly associated with different microclimates [21]. In Reunion

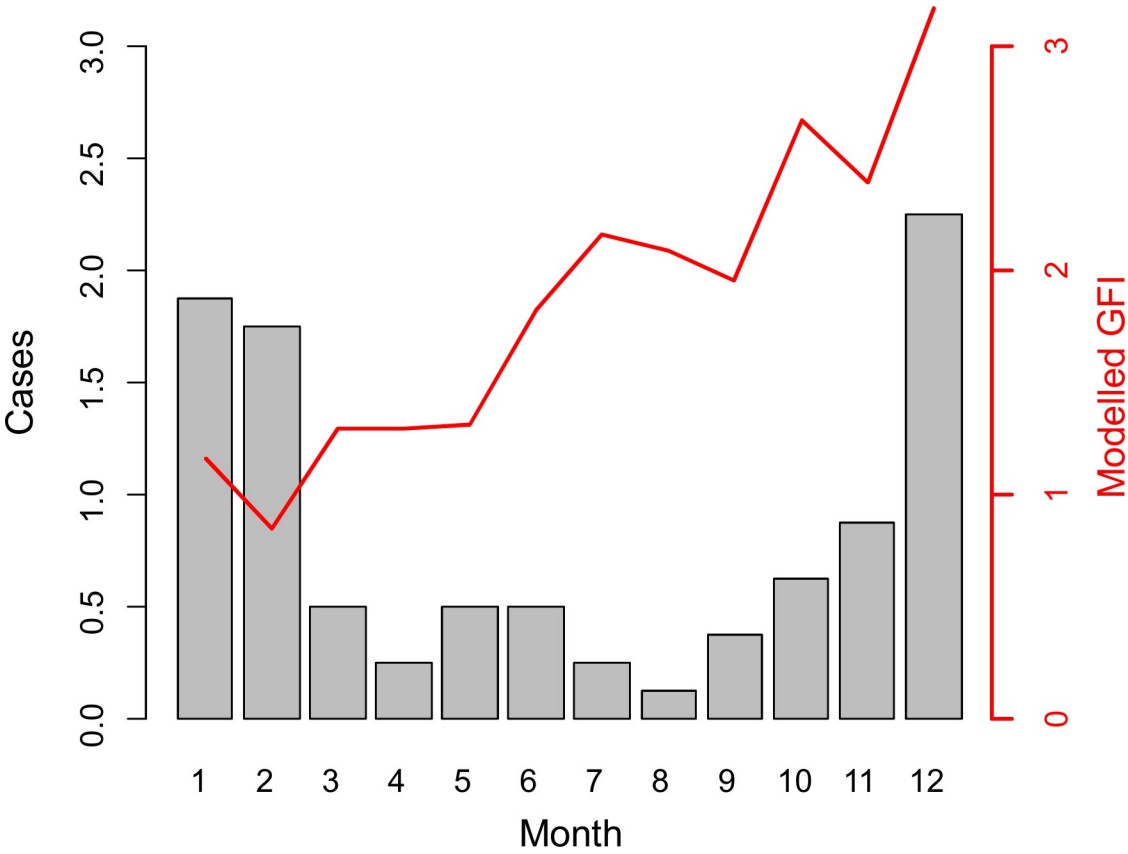

**Fig 5. Annual predicted dynamics of the *Xenopsylla* Genus Flea Index (GFI) according to temperature and rainfall.** The red line represents the modelled GFI from meteorological data collected at Pierrefonds weather station, Reunion Island, 2018. The grey bars correspond to the number of murine typhus cases averaged over the 2011–2019 period.

Island, heavy rainfall and the architecture of burrows in the windward coast may be unsuitable to the development of *Xenopsylla* spp. and hence lead to a virtual absence of these vectors on the East side of the island, whereas higher temperatures favor the development of *Xenopsylla* spp. in the West side of the island.

According to our results, different land cover types also impacted the *Xenopsylla* GFI, namely the land cover classes "discontinuous urban", "forest", "herbaceous vegetation" and "agriculture", all positively associated with rat flea abundance. This result suggests that landscapes favorable to *Xenopsylla* fleas on Reunion Island are preferentially located in suburban or rural areas. Moreover, such landscapes are generally fragmented and composed of a mix of land covers, including residential, agricultural and natural areas. The role of land use and land cover on flea indices was also demonstrated in previous studies conducted in Lushoto District, Tanzania, revealing that fallow land and natural forest had higher flea indices [22]. Of note, the land cover classes "continuous urban", and "water" were both negatively correlated with rat flea abundance in the univariate analysis (S4 Table). On the other hand, the land cover classes "shrub vegetation" and "barren land" were not retained in the multivariate model, although they were positively and significantly correlated with rat flea abundance in the univariate analysis (S4 Table). These results stress the need for additional sampling studies to decipher the role of landscape elements in rat flea abundances in Reunion Island.

The developed predictive map of *Xenopsylla* fleas significantly overlaps that of murine typhus human incident cases (Fig 6A), suggesting that risk areas for murine typhus on

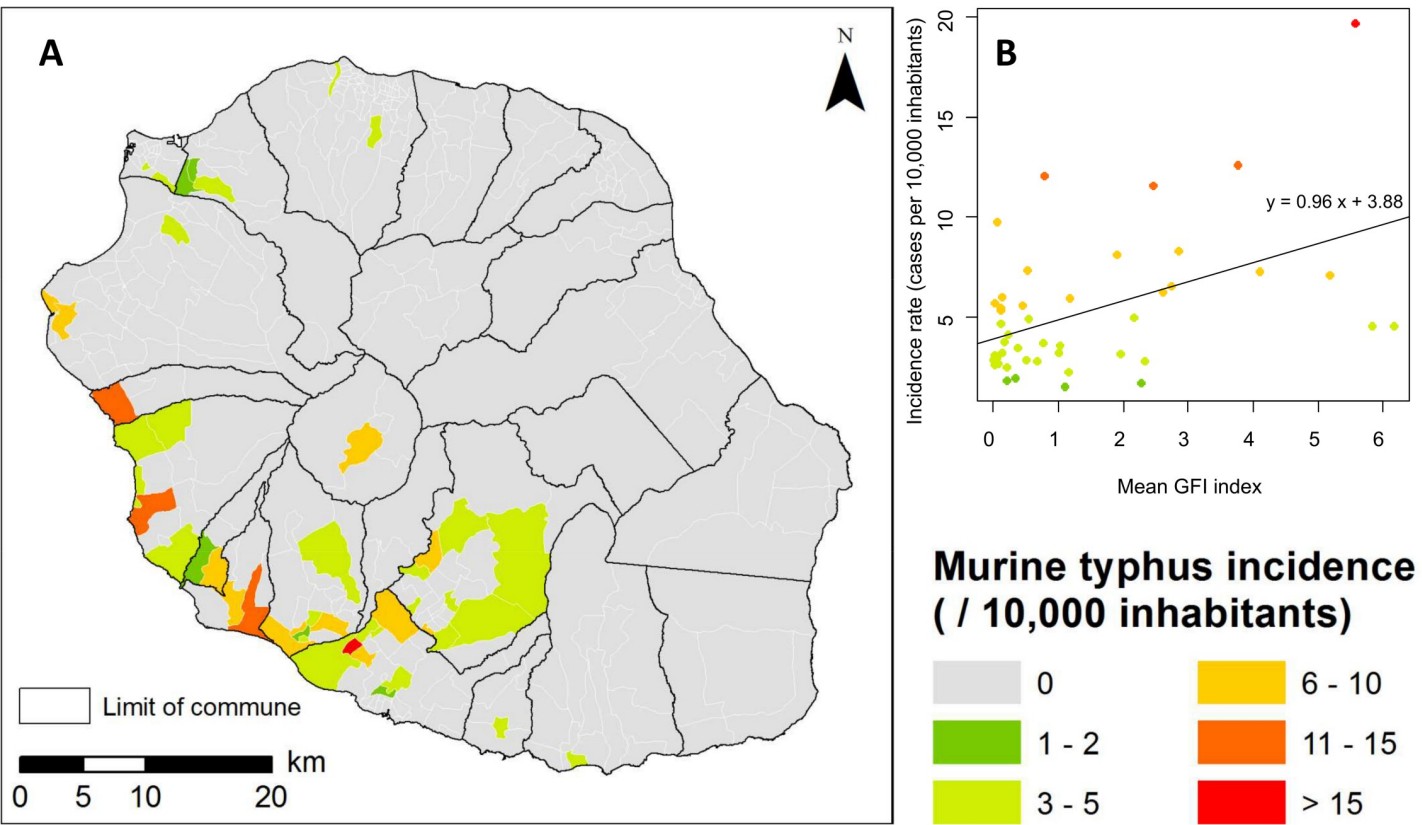

**Fig 6. A) Murine typhus incidence on Reunion Island, 2011–2018** (source for the administrative limits: French national geographic institute BD TOPO–data available under the free license Open license 2.0) **B) Bi-dimensional representation of the mean predicted GFI and observed murine typhus incidence at IRIS census level.** The black line is the regression line.

Reunion Island are mainly located in suburban or rural areas of the western leeward coast. Of note, this pattern differs from those shown in different geographic contexts: for example, a seroprevalence survey conducted in Vientiane, Lao PDR, demonstrated that murine typhus transmission occurs mainly in densely populated urban areas [23]. Indeed, the impact of abiotic factors is expected to be different in each environmental setting, notably because of the composition and abundance of the host and vector species that are locally involved in *R. typhi* transmission.

### Temporal dynamics of flea infestation

In our study, a 12-month trapping design was set up to investigate the temporal dynamics of rat fleas. The measured *Xenopsylla* GFI reflects the flea infestation of the rodent population throughout the year. Our analysis highlighted GFI variations over time in relation with temperature and rainfall: hot (daily maximum temperature above 27˚C) and dry (maximum rainfall over the last 42 days below 60 mm) conditions favor the infestation of rats (Fig 4). This result is in keeping with the spatial distribution of *Xenopsylla* fleas modeled using a distinct dataset and showing highest infestation in the hot and dry regions of the island (Fig 3).

There have been few investigations of flea abundance in tropical environments [24–26]. Makundi and co-workers [24] documented a strong seasonality in flea infestation in Tanzania with a marked peak in December that mirrors figures presented herein, although Tanzania and Reunion Island are obviously different environmentally. An older survey also reported

sharp but asynchronous peaks in *X. cheopis* and *S. fonquiernei* abundance on rats trapped in Tananarive, Madagascar [26]. These pioneering investigations stress the need for studies of the biological cycles of fleas, including that of off-host stages that remain dramatically overlooked. It is also of utmost importance to decipher the fate of adult fleas after the seasonal peak of abundance. Some previous experimental studies have revealed strong but complex responses of adult rat fleas in response to relative humidity. Indeed, unfed adult fleas were shown to aggregate at the dry side of experimental chambers when exposed to increasing relative humidity [27]. In addition, the activity (number of moving, crowling and jumping fleas) of adult unfed fleas was strongly positively correlated with increasing relative humidity [27].

We can hypothesize that, following abundance peak, fleas leave their hosts to reproduce, or that increasing humidity and temperature at the beginning of the humid season favors rat-human host shift as a consequence of increased activity of fleas, as suggested by experimental data on other *Xenopsylla* species [27]. Although the biological mechanisms underlying these dynamics are unknown, it is important to note that peaks of flea infestation and human cases of murine typhus examined herein nearly coincide in December (Fig 5). Therefore, understanding the biological cycle of rat fleas at this period is pivotal and warrants further investigation.

## Consequences for human disease control recommendations

The predictive map of flea infestation (Fig 3) and the seasonality of flea infestation should both be considered for targeted rat control programs on Reunion Island. Indeed, the predictive map shows that only a limited portion of the island is actually home to a significant abundance of rat fleas. The seasonality of infestation further suggests that, in highly infested areas, flea infestation may be of concern mainly during the first months of the hot humid season. Rat control is carried out throughout the island and year-round to protect farms and crops, as well as vulnerable flora and fauna of the island. In addition to town halls, two regional services are in charge of rodents control in sugar cane fields, and in cattle and pig farms, using rodenticide. In the absence of associated control of fleas, rodenticides have a limited impact on fleas [28] and rat control may actually increase the exposure of human populations to these invertebrate vectors. For this reason, we propose the use of Kartman boxes–a method associating an anticoagulant rodenticide and an insecticide–as recently tested in Madagascar [28], and this only in areas and seasons at risk, *i.e.* in the West and South portions of the island at the end of the dry season.

With regard to public health, these findings suggest that family doctors located in the affected areas of the south and the west should be informed regarding murine typhus risk as well as diagnosis and treatment before the annual high-risk period. Hygiene promotion and information on the disease transmission together with advice to protect pets from flea infestation could also be offered to the population living in these affected areas.

## Limitations and perspectives

In this study, we investigated the impact of environmental and climatic drivers on the abundance of adult rat fleas on Reunion Island, and our results showed significant relationships between environmental conditions, the abundance of rat fleas, and the incidence of murine typhus in humans. Published data showed that the occurrence of rat fleas is strongly structured geographically on Reunion Island [12], which oriented our study towards the exploration of invertebrate vectors. However, both analyzed datasets only included outdoor trapping results [12]. Indoor trapping of rats and rat fleas would be necessary to complement our study. The role of *Ctenocephalides felis*–the cat flea–as a vector of *R. typhi* should also be investigated [29].

Moreover, other non-exclusive hypotheses could be examined in the future, especially the temporal dynamics of infection in fleas (H3) and of the exposure of humans to rats (H4). Measuring *R. typhi* prevalence in rat fleas throughout the year will certainly complement our understanding of the dynamics of human disease. Addressing the temporal variations in rat abundance could be achieved through capture-marking-recapture experiments [17], which could be implemented specifically in the sites with highest risk, *i.e.* sites with highest human seroprevalence or, in the absence of such data, in sites with dense human population and highest estimated GFI. A seasonal increase in human exposure to rats may be concomitant with the harvest of sugar cane (June-December), during which rats are forced to disperse and possibly seek shelter in or near human habitats.

Finally, it should be noted that Reunion Island can be considered as a simplified environmental setting, as it is a relatively closed system sheltering only two rat species, *R. rattus* and *R. norvegicus*, together with two rat flea species competent for *R. typhi* transmission, namely *X. cheopis* and *X. brasiliensis*. In comparison, the neighboring island of Madagascar shelters almost 30 species of endemic rodents and an even higher number of endemic fleas [30,31], in addition to the introduced rodent and flea species that are common to both islands. Thus, the extrapolation of study results obtained on Reunion Island would certainly not be relevant for other Indian Ocean islands. However, the methods used here, combining GIS and statistical inference approaches to explore the different links between environment, reservoirs, vectors and human hosts, could be applied to other geographic contexts.

## Supporting information

**S1 Table. Location of the study sites.**
(PDF)

**S2 Table. Dataset 1.**
(PDF)

**S3 Table. Dataset 2.**
(PDF)

**S4 Table. Results of univariate analysis.**
(PDF)

**S5 Table. Results of SVM analysis: list of the ten models with the best performance in terms of mean square error.** *TMAX_N* denotes the maximum temperature over the last *N* days, *TMIN_N* denotes the minimum temperature over the last *N* days, *RAINMAX_N* denotes the maximum rainfall over the last *N* days, and *RAIN_N* denotes the rain accumulation over the last *N* days. Models are ordered from best to worst.
(PDF)

**S1 Text. Trapping protocol, Dataset 2.**
(PDF)

## Acknowledgments

The authors thank Dr Anne Laudisoit for her help on the fleas' morphological diagnosis.

## Author Contributions

**Conceptualization:** Annelise Tran, Patrick Mavingui, Eric Cardinale, Pablo Tortosa.

**Data curation:** Annelise Tran, Gildas Le Minter, Elsa Balleydier, Vanina Guernier, Erwan Lagadec.

**Formal analysis:** Annelise Tran, Elsa Balleydier.

**Funding acquisition:** Patrick Mavingui, Eric Cardinale.

**Investigation:** Gildas Le Minter, Anaïs Etheves, Morgane Laval, Floriane Boucher, Vanina Guernier, Erwan Lagadec.

**Methodology:** Annelise Tran, Pablo Tortosa.

**Project administration:** Pablo Tortosa.

**Resources:** Annelise Tran.

**Software:** Annelise Tran.

**Supervision:** Annelise Tran, Pablo Tortosa.

**Validation:** Annelise Tran.

**Visualization:** Annelise Tran.

**Writing – original draft:** Annelise Tran, Pablo Tortosa.

**Writing – review & editing:** Annelise Tran, Gildas Le Minter, Elsa Balleydier, Anaïs Etheves, Morgane Laval, Floriane Boucher, Vanina Guernier, Erwan Lagadec, Patrick Mavingui, Eric Cardinale, Pablo Tortosa.

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
