## [Decision Letter · Decision Letter 0]

14 Oct 2020

Dear Dr. Tran,

Thank you very much for submitting your manuscript "Describing fine spatiotemporal dynamics of rat fleas in an insular ecosystem enlightens abiotic drivers of murine typhus incidence in humans" for consideration at PLOS Neglected Tropical Diseases. As with all papers reviewed by the journal, your manuscript was reviewed by members of the editorial board and by several independent reviewers. The reviewers appreciated the attention to an important topic. Based on the reviews, we are likely to accept this manuscript for publication, providing that you modify the manuscript according to the review recommendations. 

Sincerely,

Peter C. Melby, M.D.

Associate Editor

Richard Phillips

Deputy Editor

Reviewer's Responses to Questions

**Key Review Criteria Required for Acceptance?**

**Methods**

-Are the objectives of the study clearly articulated with a clear testable hypothesis stated?

-Is the study design appropriate to address the stated objectives?

-Is the population clearly described and appropriate for the hypothesis being tested?

-Is the sample size sufficient to ensure adequate power to address the hypothesis being tested?

-Were correct statistical analysis used to support conclusions?

-Are there concerns about ethical or regulatory requirements being met?

Reviewer #1: The objectives of the study and hypotheses being tested were clearly articulated. The study design was appropriate with sufficient sample size and a clear description of the population. The statistical analyses were appropriate and I have no concerns regarding ethical or regulatory requirements.

Reviewer #2: The methods are clearly written and seem appropriate to address the authors' questions.

Reviewer #3: Sound and well described methodology. Sample size rather small but analyses were directed towards different outcomes. Study limitations acknowledged. Ethical statement appropriate.

**Results**

-Does the analysis presented match the analysis plan?

-Are the results clearly and completely presented?

-Are the figures (Tables, Images) of sufficient quality for clarity?

Reviewer #1: The results were presented clearly and figures are of sufficient quality.

Reviewer #2: The data is presented clearly.

Reviewer #3: Clearly depicted results. Figures and Tables of sufficient quality and covering study outcomes.

**Conclusions**

-Are the conclusions supported by the data presented?

-Are the limitations of analysis clearly described?

-Do the authors discuss how these data can be helpful to advance our understanding of the topic under study?

-Is public health relevance addressed?

Reviewer #1: The conclusions and the limitations of the data are clearly presented and discussed. Both the public health importance of the findings and broader implications of the study are discussed.

Reviewer #2: The conclusions are reasonable based on the presented data and have public health relevance. The discussion highlights the limitations of the study in a balanced manner.

Reviewer #3: Conclusions supported by data and results and corresponding to study aims. Well written discussion with appropriate literature and comparisons. Public health relevance is clear throughout the manuscript.

**Editorial and Data Presentation Modifications?**

Reviewer #1: I found the choice of words odd and awkward in a few places, and specific suggestions for more appropriate word choices and sentence structure are listed below. 

Specific suggestions:

Line 27: change structuration to structure, delete "incident" before cases

Line 29: change "compartment" to "component"

Line 30: change "of" to "from"

Line 49; awkward sentence: change “which number of cases” to “as the number of human cases”….

Line 52: delete "thus"

Line 67: change "highlight" to "determine"

Line 78: add comma after niche

Line 84: change "setups" to "settings"

Line 94: change "up-cited" to "previous"

Line 94: Authors state “its emergence in Reunion Island may have

 resulted from suboptimal surveillance and diagnosis prior the identification of the first

incident cases in 2012. This sentence needs to be reworded…e.g. detection of emergence may have been delayed due to suboptimal surveillance and diagnosis prior to identification of the first human cases in 2012. 

Line 97: disease should be diseases

Line 100: structuration is a strange word, replace with structure here and throughout the paper

Line 101: add "with" after seasonality, and before cases

Line 104: “it typically is” should be “as is typical”

Line 118: change "up-cited" to "previously stated"

Line 120: incident can be deleted…human cases is sufficient

Line 124: "aiming" should be "aimed"

Line 133; insert comma after surveillance, delete “an access to”

Line 134; insert “access” before “to health data”

Lines 135 and 136: replace “run” with “conducted according to”

Line 155: replace “a” with “rodent”

Line 169: add “in humans” after cases, reemphasize throughout that you are referring to incidence or cases of human typhus

Line 259: Univariable should be univariate here and elsewhere

Line 260: multivariable should be multivariate here and elsewhere

Line 320: replace “leading to” with “and a” (non specific symptoms don’t lead to lack of awareness but the two combined lead to underestimation)

Line 323: replace “explored rodents and rodent fleas compartments” with “ analyzed factors related to rodents and rodent fleas, …. 

Line 324: replace “the presented analysis enlightens” with “ The results of our analysis demonstrates that climatic and environmental variables strongly affect….

Line 327: replace structuration with "occurrence"

Line 331: flea instead of fleas

Line 340: burrow instead of burrows before architecture

Line 341: delete "burrows"

Line 376: delete "somehow" 

Line 377: delete set-ups. Change environment to environmentally. 

Line 380: change “an exploration” to “studies” 

Line 381: Not sure what pre-imaginal means, use off-host or questing fleas

Line 382: Not sure what after the “seasonal pick” means…please clarify

Line 407: "theses" should be "these"

Line 437: set-up should be "setting"

Line 442: delete "this"

Reviewer #2: (No Response)

Reviewer #3: Accept

**Summary and General Comments**

Reviewer #1: In general, this is an interesting and well-organized study on factors associated with rats and rat fleas and the incidence of human typhus on Reunion Island. The authors did a nice job describing the problem, their study design and their findings and relating that to broader human health implications.

Reviewer #2: The manuscript by Tran and colleagues describes the environmental conditions that appear favorable to Xenopsylla infestation of rats on Reunion Island, where murine typhus has emerged as a cause of illness – especially in the dryer areas of the south and western parts of the island. On a local level (i.e., Reunion Island) the results can have an impact on control efforts. Of broader interest, the authors’ approach could help others postulate how to understand and approach factors associated with R. typhi transmission in other regions. Below are some minor suggestions and queries for the authors’ consideration. 

Line 55: Suggest qualifying the phrase “main determinants” as of those investigated within the confines of this study. 

Lines 87 – 94: The details regarding the emergence of chikungunya, dengue, and how they are different than that of R. typhi is not necessary for an already lengthy introduction. 

Line 97: Suggest finding an alternate phrase for “off the radar.” 

Line 104: Provide citation(s) (additional to #10) to support that rats are dispersed in a widespread manner on tropical islands. 

Line 177 – 178: How is the serologic confirmation by the reference center different than the previously stated serologic criteria? If different, the methods or criteria should be stated. If the same, the reference center can be mentioned as a resource in the text but not a specific criterion in the bulleted list. 

Line 229: Very minor: The phrase “GFI index” seems redundant. Would just “GFI” suffice? 

Line 297 – 298: Can the authors provide the length of lag time between rapid decrease in GFI and the eventual decrease in human cases? Although it might be expected that there might be a lag, this might be counter intuitive to some readers. The lag time length (if relatively short) might help put things into more plausible context. 

Lines 346 – 354: Can the authors’ compare and contrast the types of land cover classes here (those with impact versus those without impact) and offer hypotheses as to why they are different? Although some differences in regard to presence of reservoirs/vectors seem obvious (i.e., continuous urban and water), others are not (e.g., shrub vegetation versus herbaceous vegetation).

Reviewer #3: (No Response)

PLOS authors have the option to publish the peer review history of their article (what does this mean?). If published, this will include your full peer review and any attached files.

Reviewer #1: No

Reviewer #2: No

Reviewer #3: No
---

## [Editor Report · Decision Letter 1]

2 Dec 2020

Dear Dr. Tran,

We are pleased to inform you that your manuscript 'Describing fine spatiotemporal dynamics of rat fleas in an insular ecosystem enlightens abiotic drivers of murine typhus incidence in humans' has been provisionally accepted for publication in PLOS Neglected Tropical Diseases.

Best regards,

Peter C. Melby, M.D.

Associate Editor

Richard Phillips

Deputy Editor

None

---

## [Editor Report · Acceptance letter]

10 Feb 2021

Dear Dr. Tran,

We are delighted to inform you that your manuscript, "Describing fine spatiotemporal dynamics of rat fleas in an insular ecosystem enlightens abiotic drivers of murine typhus incidence in humans," has been formally accepted for publication in PLOS Neglected Tropical Diseases.

Best regards,

Shaden Kamhawi

co-Editor-in-Chief

Paul Brindley

co-Editor-in-Chief
